

# Synchrony on the reef: how environmental factors shape coral spawning patterns in *Acropora* corals in the Maldives

Kate Sheridan[1,*], Margaux A.A. Monfared[1,2,*], Simon P. Dixon[1,3], Amelia J.F. Errington[1] and Thomas Le Berre[1]

[1] Reefscapers Pvt Ltd, Male, Maldives
[2] Blue Pangolin Consulting Ltd, London, United Kingdom
[3] Coral Vita, Nassau, Bahamas
[*] These authors contributed equally to this work.

## ABSTRACT

Elucidating our knowledge on the reproductive phenology of scleractinian corals and the environmental drivers of reproductive synchronicity is pivotal for assessing gene flow between populations and the potential for ecosystem recovery. The timing of gamete release in sessile broadcast spawning corals is key to successful reproduction; and is dependent on a complex interaction between an organism's genes and external environmental factors. In this study we assessed the effect of various environmental factors on the spawning timing and synchronicity of *Acropora* corals in the Maldives. A total of 3,026 colonies from 24 species of *Acropora* were recorded spawning between October 2021 and May 2024: 1,709 from 20 species in North Male Atoll and 1,317 from 18 species in Baa Atoll. Generalised linear models (GLMs) were used to estimate the effect of average daily wind speed (mph), total daily precipitation (mm), tide depth (m), and mean sea surface temperature (SST) over a 30-day period prior to spawning, on the proportion of colonies to spawn per day and the spawning day deviation to full moon. Models were run for all *Acropora* corals, and three species in which more than 30 days of spawning were observed enabling robust statistical models to be tested: *A. secale*, *A. tenuis*, and *A. humilis*, to determine the presence of species-specific relationships. Based on additional GLMs, we found that a change in SST does not determine the likelihood of *Acropora* spawning to occur in a given month, but does significantly predict the number of *Acropora* colonies to spawn per month. We also found that the relationship between SST and spawning as a predictor of probability or synchronicity on a monthly temporal scale can be species specific. We found a significant, positive correlation between daily precipitation levels and the proportion of *Acropora* colonies to spawn per day, however, there were some variations between species. Additionally, a higher proportion of *Acropora* colonies spawned closer to the full moon. Spawning events of *Acropora* corals closer to the full moon are significantly correlated with lower tide depths across both atolls. This knowledge will be beneficial for the management of reef systems in the Maldives following a global bleaching event, due to increased reliance on targeted conservation measures to retain diversity and re-populate degraded reefs, such as *in-situ* larval settlement. While our analyses of environmental factors goes someway in explaining variability in spawning patterns within the *Acropora* genus in the Maldives, we must also conclude there are other factors which remain unexplored, or there is a wide range of ecologically appropriate conditions for spawning. However,

Corresponding authors
Kate Sheridan,
sheridankate0@gmail.com
Margaux A.A. Monfared,
margauxmonfared@yahoo.com

our results highlight the importance of considering environmental conditions, and species-specific relationships, when predicting *Acropora* spawning, due to the temporal and spatial deviations in timing and synchronicity observed within and between species.

## INTRODUCTION

Scleractinian corals are autogenic ecosystem engineers that create habitat spaces for a variety of marine organisms and are key drivers of diversity in reef ecosystems (*Jones, Lawton & Shachak, 1994*; *Wild et al., 2011*). As such, reproduction of reef-building corals holds paramount ecological significance and is fundamentally important for elucidating our knowledge on reef ecology and evolution. Through broadcast spawning events, in which gamete bundles are released into the water column by different colonies, the adaptability and resilience of coral populations to environmental changes is enhanced. This mechanism of sexual reproduction increases genetic diversity and is critical for coral reef recovery following mass mortality (*Gilmour et al., 2013*; *Baums et al., 2013*; *Davies et al., 2023*). Additionally, coral spawning provides a substantial influx of organic material into the water, serving as a valuable food source for marine organisms and contributing to the productivity of the wider reef ecosystem (*Westneat & Resing, 1988*; *McCormick, 2003*; *Wild et al., 2008*). Therefore, monitoring coral spawning events serves as a valuable indicator of the state of coral reefs at an individual and assemblage level and proves essential in evaluating and addressing the impacts of climate change on coral reef ecosystems.

The timing and synchronicity of coral spawning can vary within and between species, spatially between reef systems, and temporally throughout the year and from year to year. Brooding scleractinian coral species are known to exhibit a relationship between lunar phase and planulae release (*Szmant-Froelich, Reutter & Riggs, 1985*). Synchronous mass spawning events have been well documented on the Great Barrier Reef in the days following the full moon in October, November and December (*Harrison et al., 1984*; *Willis et al., 1985*; *Babcock et al., 1986*), whereas temporal reproductive isolation was noted for corals in the Red Sea, with spawning of 12 of 13 studied species occurring in different seasons, months or lunar phases (*Shlesinger & Loya, 1985*). Between these extremes, asynchronous spawning, extended reproductive periods and spawning patterns that differ significantly between taxa have been widely reported in equatorial regions (*Oliver et al., 1988*; *Mangubhai & Harrison, 2008*; *Baird, Guest & Willis, 2009*; *Isomura & Fukami, 2018*; *Gouezo et al., 2020*). Equatorial regions experience weaker environmental seasonality than higher latitude environments, and it has been hypothesised that this enables multiple spawning events a year and results in a breakdown in spawning synchrony at lower latitudes (*Oliver et al., 1988*). However, many studies document multi-specific spawning in equatorial regions, challenging this hypothesis (*Penland et al., 2004*; *Guest et al., 2005a*; *Sola, Marques Da Silva & Glassom, 2016*; *Wijayanti et al., 2019*). These challenges have led
to the argument that no coastal environment is 'aseasonal' (*Guest et al., 2005b*). Thus, seasonal cues, invoked by environmental conditions, such as weather conditions, tidal patterns, sea temperature fluctuations, and lunar cycles, likely play a role in regulating coral reproductive timing. On Palauan reefs, *Gouezo et al. (2020)* noted differences in spawning months and environmental drivers of spawning between genera, as well as inter- and intraspecific spawning asynchrony within taxa. Further, *Mangubhai & Harrison (2008)* reported highly asynchronous spawning patterns for *Acropora* species on Kenyan reefs, with peak spawning month (*i.e.*, the month with the highest number of colonies to spawn) differing between species, and for some species peak spawn month differed between years. Their results showed that whilst almost all colonies had a single annual cycle of gametogenesis, some colonies of *A. valida* had bi-annual cycles. Bi-annual spawning cycles have also been reported in colonies of *A. humilis* off Singapore (*Guest et al., 2005b*) and in *Acropora* corals off western Australia (*Rosser & Gilmour, 2008*). Additionally, species-specific spawning patterns have been documented within the *Acropora* genus on the Great Barrier Reef, with some species spawning during synchronous mass spawning events alongside other genera and others spawning at additional times (*Wallace, 1985*). Understanding the environmental factors that drive variations in reproductive cycles within and between species across a range of geographic locations is key for assessing changes in spawning synchrony and reproductive success under rapidly changing climatic conditions.

Local and regional environmental conditions have been linked to spawning patterns across varying temporal scales. The exact timing of gamete release in sessile broadcast spawning corals is key to successful reproduction; and is dependent on a complex interaction between an organism's genes and external environmental factors, which may act as ultimate or proximate cues (*Forrest & Miller-Rushing, 2010*). Corals have therefore evolved to spawn during optimal conditions (*Foster, Heyward & Gilmour, 2018*), with synchronous gamete release favoured to maximise cross fertilisation between colonies and augment reef connectivity (*Levitan et al., 2004*; *Guest, 2008*; *Nakajima et al., 2010*; *Romero-Torres, Acosta & Treml, 2017*; *Van Der Ven, Ratsimbazafy & Kochzius, 2022*). With emerging restorative methods utilising sexual reproduction to replenish degraded reefs (*Randall et al., 2020*; *Harrison, 2024*), determining patterns of coral spawning at local and regional scales is essential for active reef restoration. Elucidating our knowledge on the reproductive phenology of scleractinian corals and the environmental drivers of reproductive synchronicity is pivotal for assessing gene flow between populations and the potential for recovery under changing climatic conditions, and can inform management of both restoration initiatives and protective measures (*Graham, Nash & Kool, 2011*; *Johnston et al., 2020*).

The exact timing of gamete release on the night of spawning has been linked to the timing of the sunset and the low tide (*Babcock et al., 1986*). When considering the night of spawning, the effect of the lunar cycle has been widely reported (*Sakai et al., 2020*). While coral spawning has been recorded over multiple lunar phases, mass coral spawning events on multiple tropical reef systems have been consistently documented within a few days of a full moon (*Hayashibara et al., 1993*; *Sakai et al., 2020*; *Lin et al., 2021*; *Komoto et al., 2023*; *Monfared et al., 2023*). A direct positive correlation between the duration of regional calm

periods and the month of mass spawning events was reported by *Van Woesik (2010)*, and wind speeds have been weakly correlated with spawning month whereby coral spawning is more probable during periods of calm-to-moderate winds (*Keith et al., 2016*). Wind speed has also been linked to deviations in spawning day relative to the full moon as *Sakai et al. (2020)* found low wind speeds to be correlated with earlier spawning days relative to the full moon. Low-to-moderate wind speeds during coral spawning events can provide evolutionary advantages, such as aiding fertilisation by reducing rapid dilution of gametes, facilitating larval retention, and encouraging local recruitment (*Van Woesik, 2010*).

The effect of rainfall on spawning date is unclear. *Monfared et al. (2023)* reported that earlier spawning dates relative to the full moon are significantly correlated with lower daily precipitation, supporting an interaction noted by *Mendes & Woodley (2002)* between monthly mean rainfall, spawning time, and temperature. High levels of precipitation decrease water salinity, which can be detrimental to the reproductive success of corals (*Scott, Harrison & Brooks, 2013*). However, there is limited evidence as to whether variations in daily rainfall influence the night of spawning (*Keith et al., 2016*; *Sakai et al., 2020*).

Contrastingly, sea surface temperatures (SST) have been widely reported as a key driver of coral spawning across various temporal scales (*Lin & Nozawa, 2023*). Rapid rises in SST have been correlated with spawning of *Acropora* in the Indo-Pacific across a range of latitudes (*Keith et al., 2016*; *Novriansyah et al., 2023*), suggesting this could be a driver of synchronicity. Similarly, rapid rises in SST have been shown to accelerate the date of spawning relative to the full moon by advancing gamete maturation and as a proximate cue for spawning events (*Nozawa, 2012*; *Keith et al., 2016*; *Sakai et al., 2020*; *Lin & Nozawa, 2023*; *Osman et al., 2024*). *Mangubhai & Harrison (2008)* found that most *Acropora* spawned when mean SSTs were at their annual summer maximum, and 42% of colonies spawned during rising SSTs. However, on Palauan coral reefs, spawning has been documented during both rising and falling SSTs, and multiple spawning events are better predicted by the rise and fall of solar insolation (*Penland et al., 2004*). Interestingly, *Gouezo et al. (2020)* found that whilst change in monthly SST was positively correlated with *in situ* solar insolation, neither variable was related to spawning of *Acropora* on Palauan reefs; spawning was more strongly associated with a monthly mean SST of approximately 29 °C and lowest yearly high tides. More studies on equatorial reefs are key for assessing the extent to which changes in SST, and monthly mean SSTs drive spawning synchrony across geographic regions.

Maldivian coral reefs extend between latitudes of 07°06′30″N and 00°41′48″S and to date, knowledge on their reproductive phenology, the seventh largest reef system globally (*Dhunya, Huang & Aslam, 2017*), is limited. *Monfared et al. (2023)* documented spawning patterns of *Acropora* corals from two Maldivian reef systems of both asexually propagated and naturally occurring colonies. This research identified two peak spawning seasons per year: in March/April and from October to December, coinciding with the change in monsoons. Similar seasonal patterns have been observed in Australia (*Rosser & Gilmour, 2008*; *Gilmour, Speed & Babcock, 2016*), Sri Lanka (*Kumara, Cumaranatunga & Souter, 2007*), Singapore (*Guest, 2005*), and Indonesia (*Permata et al., 2012*; *Wijayanti et al., 2019*; *Novriansyah et al., 2023*). *Monfared et al. (2023)* found inter-seasonal differences in species,

timing, and synchronicity, as well as inter-atoll and inter-annual variations in spawning patterns. This study also documented earlier spawning events, relative to the full moon day, were correlated with lower tide depths, average daily wind speeds, total daily precipitation levels, and higher daily SSTs. However, how and the extent to which these environmental factors influence deviations in the synchronicity and timing of coral spawning remain unexplored.

In this study we aimed to assess the effect of environmental factors on the spawning of Maldivian *Acropora* corals. Given that environmental factors influence coral spawning on a variety of temporal scales, we sought to investigate the relationships between environmental factors and *Acropora* spawning patterns by testing three distinct hypotheses. All three hypotheses were tested for the *Acropora* genus, and individually for three species of *Acropora: A. secale, A. tenuis* and *A. humilis*, to determine if any species-specific relationships were observed. These species were selected because more than 300 colonies of each species were recorded spawning and spawning was observed over 30 days or more during the study period.

Our first hypothesis is that SSTs influence the monthly spawning patterns of *Acropora* corals. We sought to determine whether the change in SST in the month prior to spawning, or an absolute value of mean monthly SST were a stronger predictor of (i) the probability of spawning taking place in a given month; and (ii) the degree of synchronicity per month (*i.e.*, how many colonies spawn in a given month). Our second hypothesis is that the day of spawning relative to the nearest full moon within a given month is influenced by environmental conditions. This hypothesis predicts that (i) warmer SSTs in the months prior to spawning result in earlier spawning events, (ii) earlier spawning events will be correlated with lower daily wind speeds and tide depths, and (iii) higher daily levels of precipitation will delay spawning day within a month. Testing this hypothesis will help us determine how environmental factors influence the night of spawning within a lunar cycle. Finally, the third hypothesis we tested is that the number of colonies to spawn on a given day is influenced by local environmental conditions. This hypothesis predicts that a higher proportion of colonies, and thus more synchronised spawning events within a lunar phase occur on days with (i) lower mean daily wind speeds, (ii) lower daily levels of precipitation, (iii) lower tide depths on the day of spawning, and (iv) warmer average SSTs in the months prior to spawning.

## METHODS

### Study sites

Surveys took place around two resort islands in the Maldives: Landaa Giraavaru (5.2862°N, 73.1121°E) in Baa Atoll and Furana Fushi (4.2500°N, 73.5458°E) in North Male Atoll Fig. 1. Both sites participate in long-term coral restoration projects run by Reefscapers Pvt Ltd (hereafter referred to as: Reefscapers), who utilise metal frames coated in resin and sand to asexually propagate reef building corals (*Morand, Dixon & Le Berre, 2022*). The Maldives is home to 300 species of reef-building corals (*Pichon & Benzoni, 2007*). This study focused on corals within the *Acropora* genus, the most species-rich and severely affected genus by

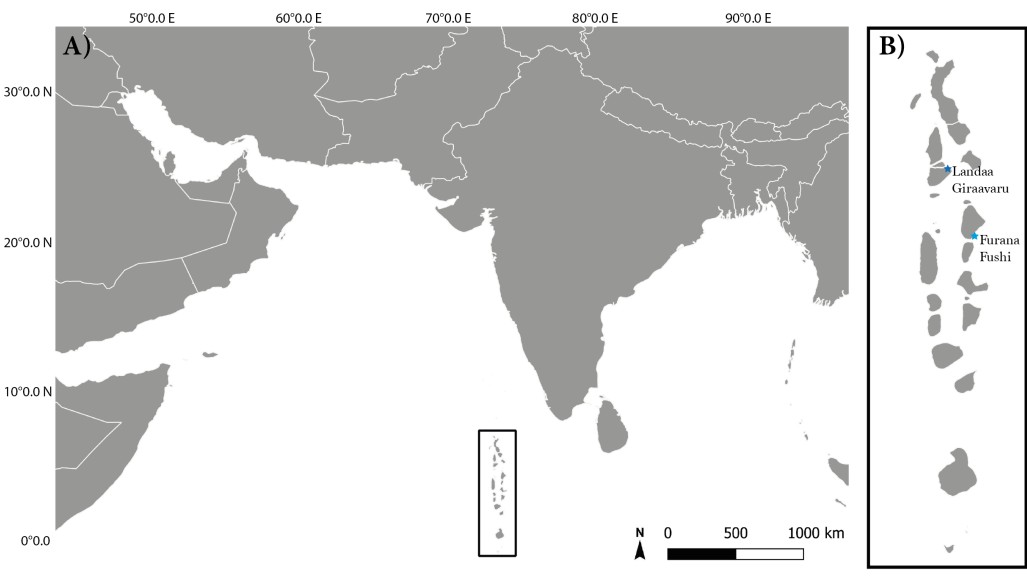

**Figure 1** **The location of the study sites in relation to equatorial region of the Indian Ocean.** (A) Map to show the latitude and longitude of the Maldivian reef system within the Indian Ocean. (B) Location of study sites in the Maldives: Landaa Giraavaru in Baa Atoll and Furana Fushi in North Male Atoll.

the 2016 coral bleaching event (*Pisapia, Burn & Pratchett, 2019*), and the focal genera of Reefscapers propagation project due to its growth rate and suitability for propagation. Survey sites were chosen with high abundance and diversity of *Acropora* species and due to accessibility for night diving or snorkelling. At Furana Fushi, data was collected from four sites: two sandy lagoons and two near-shore reef sites, at a maximum depth of 7 m. At Landaa Giraavaru, data was collected from the southern side of the island at three sites: two shallow near-shore sites, one with sand-dominated substrate and one with mixed rocky substrate as defined by *Alquezar & Boyd (2007)*, and one deeper near-shore site at a maximum depth of 10 m.

## Data collection

In this study, we used data from *Monfared et al. (2023)* (spawning observations of 1,200 colonies from 22 species of *Acropora*) collected between October 2021 and April 2023, as well as additional data (spawning observations of 1,826 colonies from 24 species of *Acropora*) collected from October 2023–April 2024. Fragmentation of mature colonies for restoration activities enabled the presence and maturity of gametes to be documented for a variety of colonies with minimal negative impacts (Fig. 2A). Upon the observation of mature gametes, based on classifications of egg maturity following *Baird, Marshall & Wolstenholme (2002)*, nightly spawning surveys took place. Nightly surveys were conducted *via* SCUBA or freediving between 16:30–22:30 from two days prior and for up to eight days after each full moon, and for two days prior for up to five days after each new moon. If spawning was observed on the final days of the scheduled survey period, additional surveys were conducted each night until a night with no spawning was observed. Surveys took place

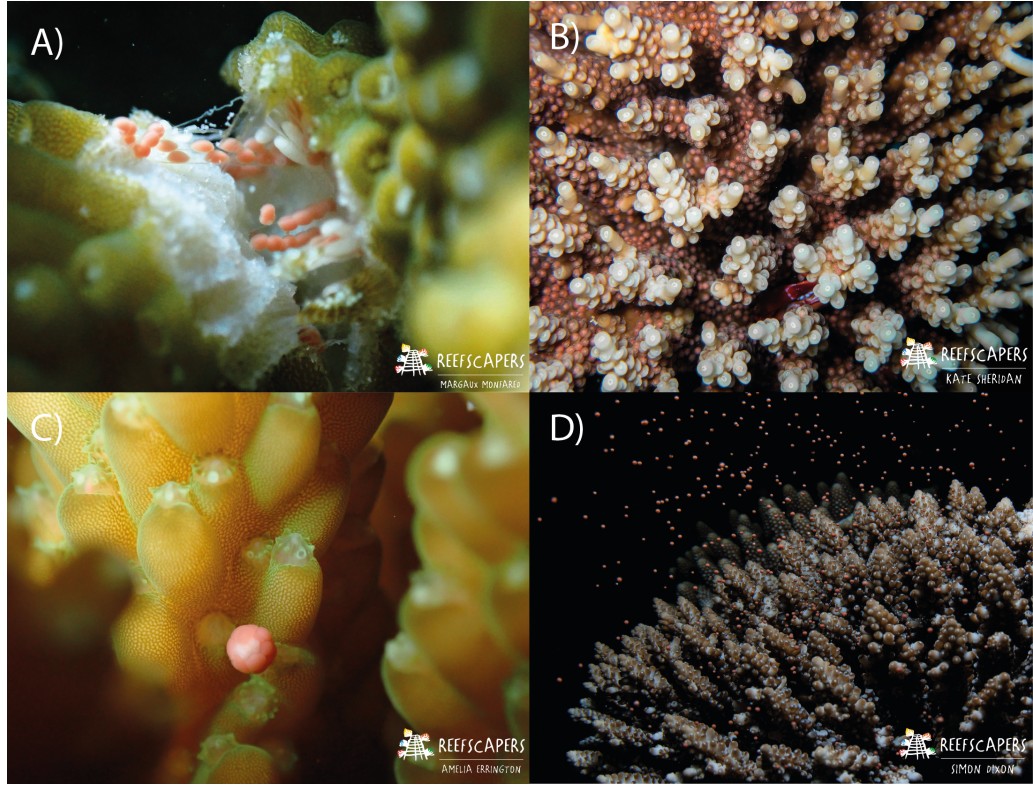

**Figure 2  Images to show the process of observing gravid colonies, setting, and spawning in *Acropora* corals.** (A) Fragmentation of *A. squarrosa* revealing gamete bundles. This process was used to identify gravid colonies. Photograph by Margaux AA Monfared. (B) Observed gamete 'setting' in *A. tenuis*. The corals hold the gamete bundles in their mouth in preparation for spawning. Observers check for this during nightly surveys. Photograph by Kate Sheridan. (C) The moment of gamete release shown in *A. secale* during a spawning event. Photograph by Amelia JF Errington. (D) Spawning of an adult colony of *A. secale*. Photograph by Simon P. Dixon.

during these lunar periods throughout March–May, and October–December each year, and additionally in months in which mature gametes had been identified (see: File S1). A minimum of two observers checked the local reef for "setting" of gravid colonies (Fig. 2B) using known setting times identified in *Monfared et al. (2023)* to accurately determine the day of spawning *via* observation of spawning onset (Fig. 2C). Individual observers remained within a specified area of the survey site throughout, both to ensure observers were within sight of each other for safety, and to ensure a maximum number of colonies could be thoroughly checked for setting. Once identified, the species, time, and location of each colony were recorded on dive slates. The same observer would repeat the checks of their area to determine spawning times. Four reef types were observed spawning: (i) wild—naturally occurring colonies, (ii) frame—asexually propagated as fragments onto Reefscapers metal frames, (iii) relocated and (iv) pyramid—colonies relocated to Furana Fushi from Gulhi Falhu, another island in North Male Atoll experiencing land reclamation. Pyramid colonies were relocated and transplanted onto metal structures, whereas relocated

colonies were directly transplanted onto the reef. For all colonies observed spawning, the date and time, reef type, species, and location was recorded.

## Environmental variables

For all colonies recorded spawning, the following environmental variables were obtained: average daily SST (°C) from seatemperatures.net (*Sea Temperature, 2024a*; *Sea Temperature, 2024b*), total daily precipitation (mm), and average daily wind speed (mph), both from *Windy.app (2024)*. These measurements were all satellite derived due to limited access to *in-situ* instruments due to funding constraints. Seatemperatures.net construct their records by combining observations from different platforms, including satellites, ships, buoys, and Argo floats, on a regular global grid before building a spatially complete SST map by interpolating to fill gaps in the data. The ECMWF (European Centre for Medium-Range Weather Forecasts) model was used to obtain precipitation and wind data from Windy.app, a weather forecast model with a 14 km resolution. Additionally, the tide depth (m) of the low tide closest to sunset was recorded, from *Tideschart (2024a)*; *Tideschart (2024b)*. Daily SSTs were recorded throughout the study period to obtain monthly averages and an average for 30 days and 90 days prior to each spawning event, in order to assess the effect of SST in the months prior.

## Statistical analyses

Generalised linear models (GLMs), with negative binomial distributions using the "MASS" package (*Venables & Ripley, 2002*), were used to test for temporal (by year), and spatial (by atoll) variations in the degree of spawning synchronicity (*i.e.*, the proportion of colonies to spawn per day, calculated as the total number of colonies to spawn per day divided by the total number of colonies to spawn), and the day of spawning relative to the full moon. A third model was used to show the proportion of colonies to spawn per day varied between days. Atoll was included as an interaction term in all models to determine if any spatial or temporal variations detected were different between the two atolls. Following these analyses, we sought to understand the effect of environmental conditions on these phenomena, and whether they can explain the temporal or geographic variations in *Acropora* spawning patterns observed in the Maldives. All statistical analyses were conducted using R Studio (*R Core Team, 2022*).

To assess whether the mean monthly SST or change in monthly SST were strong predictors of the probability of spawning to occur in a given month, we constructed a GLM fitted with a binary logistic regression. Our response variable was coded such that "1" indicates months where spawning was recorded, and "0" indicates months where no spawning activity was recorded. Secondly, to assess whether the same explanatory variables influence the degree of synchronicity per month (calculated as the total number of colonies to spawn per month), a second GLM with a negative binomial distribution was fitted. Atoll was included as a random effect in both models initially, but the variance was very low (var <0.001) indicating minimal difference between atolls. Model selection was used based on Akaike Information Criterion (AIC) and Atoll was excluded from the final model.

To determine the degree of spawning synchronicity for individual species, the proportion of colonies to spawn per day was calculated for *A. secale*, *A. tenuis*, and *A. humilis*

individually. These species-specific proportions took the total number of that species to spawn per day divided by the overall total number of colonies of that species to spawn. These proportions, as well as the proportion of all *Acropora* corals to spawn per day, were fitted as the response terms in GLMs with beta distributions using the "glmmTMB" package (*Brooks et al., 2017*). Four explanatory variables were included in the models taken for each day spawning was observed: average daily wind speed (mph), total daily precipitation (mm), tide depth at the lowest tide closest to sunset (m), and mean SST (°C) over a 30- and 90-day period prior to spawning, to cover the last stages of gametogenesis (*Sakai et al., 2020*). The day of spawning proximity to the full moon was also included as a fixed effect to determine whether more colonies spawn closer to the full moon. Explanatory variables were tested for collinearity and the two variables of SST were highly correlated ($r_{Acropora} = 0.905$, $r_{A.secale} = 0.930$, $r_{A.tenuis} = 0.911$, $r_{A.humilis} = 0.915$), and thus only mean SST over a 30-day period prior to spawning was included in the models. For all models, Year and Atoll were initially included as random effects, but neither were capturing any significant unobserved differences (var <0.001). Model selection based on AIC was conducted, and in the final models Atoll and Year were excluded as random effects.

Using the same four environmental variables as fixed effects, a GLM fitted with negative binomial distribution was used to assess their effect on spawning day. The deviation of spawning day from the nearest full moon was included as the response variable. Given previously documented differences in spawning day between atolls (*e.g.,* *Monfared et al., 2023*), Atoll was included as an interaction term with all predictors to determine whether the relationship between each environmental condition was dependent on Atoll. Month was initially included as a random effect but removed due to near-zero variance (var <0.001), and model selection based on AIC. To determine species-specific relationships between spawning day relative to the full moon and environmental conditions, separate models were built for *A. secale*, *A. tenuis*, and *A. humilis*. All response variables were checked for normality and common variance. The model of *A. secale* and *A. humilis* spawning day relative to the full moon were fitted with a Poisson distribution using the "glmmTMB" package (*Brooks et al., 2017*). The model of *A. tenuis* was fitted with a negative binomial distribution using the "MASS" package (*Venables & Ripley, 2002*). Atoll was initially included as an interaction term with all predictors in the individual species models. Interaction terms that were non-significant ($p > 0.1$) were removed through a stepwise simplification process. Final model selection was based on AIC, determining which interaction terms to retain in the final models.

For all models, the "DHARMa" package (*Hartig, 2022*) was used to assess residual diagnostics and overall model fit. Several statistical checks were performed to validate the appropriateness of the GLMs and assess key assumptions. These checks included examining simulated residual plots to test for randomness and uniformity. Additionally, a dispersion test and the sum of squared Pearson residuals were performed to check for overdispersion in the model. Residuals *versus* fitted values plots were created to check for discernible patterns. Mean and variance of the Pearson residuals were calculated to assess the variance of the residuals, and deviance residuals were examined and divided by the

degrees of freedom to examine the model fit. These diagnostics confirmed that the models were well-specified with no significant issues of overdispersion or lack of fit.

## RESULTS

A total of 3,026 colonies from 24 species of *Acropora* were recorded spawning between October 2021 and May 2024: 1,709 from 20 species in North Male Atoll and 1,317 from 18 species in Baa Atoll. Spawning was recorded in eight months of the year (see: File S2), with the greatest number of colonies recorded to spawn across both atolls in April ($N = 910$) and November ($N = 1,891$). Throughout the duration of the study, spawning was recorded on 68 days: 43 days from Baa Atoll, of which 60.5% were synchronous multispecific spawning events, defined hereafter as populations of two or more species of *Acropora* releasing gametes over the same night (*Mangubhai & Harrison, 2008*); and 38 days from North Male Atoll, of which 55.3% were synchronous multispecific spawning events (Fig. 3). In Baa Atoll the number of colonies to spawn on a given day ranged between one colony (recorded on five different days) and 203 colonies; in North Male this count ranged from one colony (recorded on 10 different days) to 207 colonies (Fig. 4). In both atolls, the day with the highest number of colonies to spawn (*i.e.*, the peak spawning day) occurred in November 2023. In Baa Atoll and North Male Atoll, 95.1% and 96.7% respectively of the colonies recorded spawning spawned during multispecific spawning events. During single species spawning events, the mean number of colonies to spawn per day was 3.82 in Baa Atoll and 3.29 in North Male Atoll.

### Temporal & spatial variations in *Acropora* spawning patterns
On days in which spawning occurred, the proximity of that day from the nearest full moon varies between atolls and Years (Fig. 5). In Baa Atoll, there is no significant difference between years in the proximity of spawning day relative to the full moon. However, in North Male Atoll, there are differences between years in spawning day relative to the full moon. Specifically, in 2024, there is a significant difference ($p = 0.006$) indicating that spawning in North Male in 2024 occurred much further from the full moon compared to the reference year 2021. Overall, spawning days in North Male are significantly closer to the full moon compared to Baa Atoll (GLM: SE = 1.159, $Z = -2.114$, $p = 0.03$). Contrastingly, on days in which spawning occurred, there is no significant difference between the two atolls in the proportion of colonies to spawn per day (GLM: SE = 0.204, $Z = -0.089$, $p = 0.929$). However, the number of colonies to spawn per day did vary significantly between days (GLM: SE < 0.001, $Z = 4.458$, $p < 0.001$), and this trend is consistent across both atolls.

### Relationship between SST and spawning month
Neither monthly mean SST or change in monthly SST were significant predictors of the probability of spawning occurring in a given month (GLM: $p = 0.517$ and $p = 0.115$ respectively, Tjur's $R^2 = 0.072$). However, change in SST was a significant predictor of the number of colonies to spawn per month ($p < 0.001$), with more colonies spawning when a rapid rise in SST occurs. When looking at a species level, monthly mean SST or

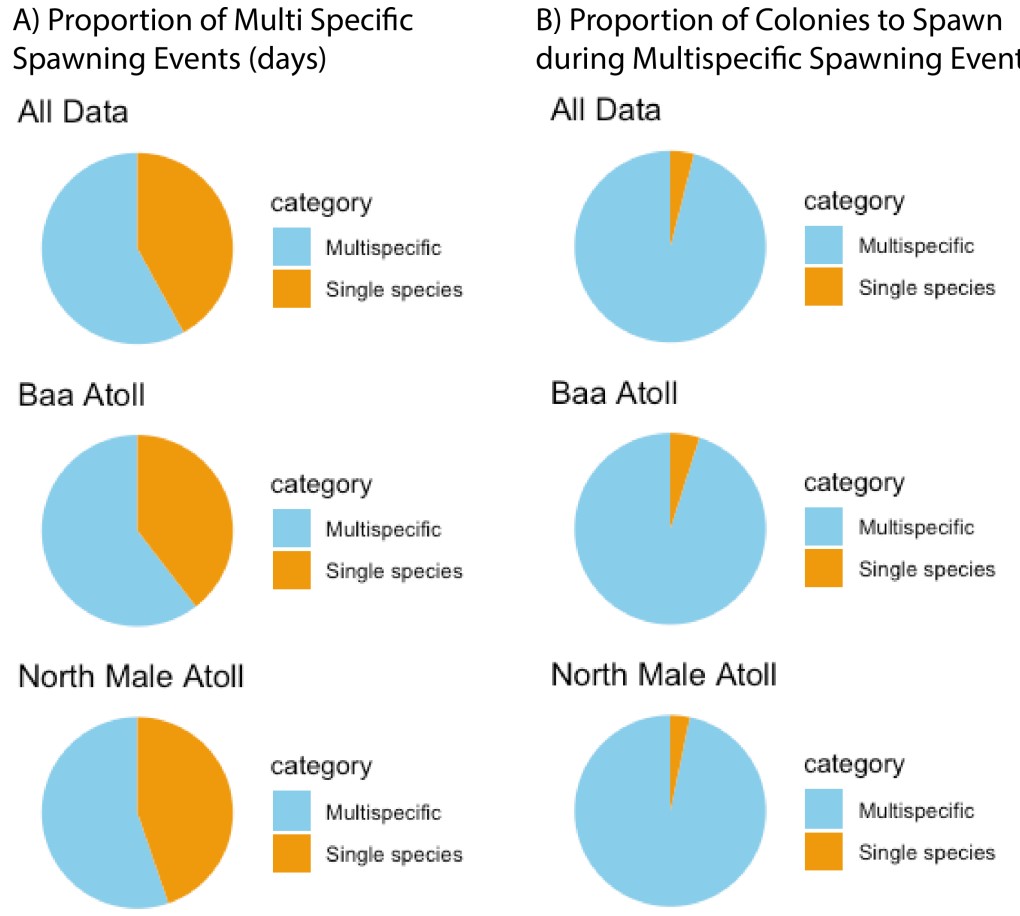

**Figure 3** **The preference of *Acropora* corals to spawn during multispecific spawning events.** (A) Proportion of spawning days ($N = 81$) recorded which were multispecific spawning events *versus* a single species. Data from all sites is combined and then separately shown for each atoll (Baa: $N = 43$; North Male: $N = 38$). (B) Proportion of *Acropora* colonies ($N = 3,026$) to spawning during multispecific spawning events *versus* a single species. Data from all sites is combined and then separately shown for each atoll (Baa: $N = 1,317$; North Male: $N = 1,709$).

change in monthly SST were not significant predictors of either the probability of spawning occurring in a given month or the number of colonies to spawn per month for *A. secale* or *A. tenuis*. However, a change in monthly SST was a significant predictor of the probability of *A. humilis* to spawn in a given month ($p = 0.007$), with spawning more likely to occur in months with higher mean SSTs. Similarly, a rise in SST was a significant predictor of the number of colonies of *A. humilis* to spawn per month ($p = 0.044$). These results indicate that the relationship between SST and spawning as a predictor of probability or synchronicity on a monthly temporal scale can be species specific.

## Effect of local environmental conditions on synchronicity

Total daily precipitation and the proximity of spawning day to the full moon were the only significant predictors of the proportion of *Acropora* colonies to spawn per day (Fig. 6). We predicted that more colonies would spawn on days with less precipitation. Instead, we

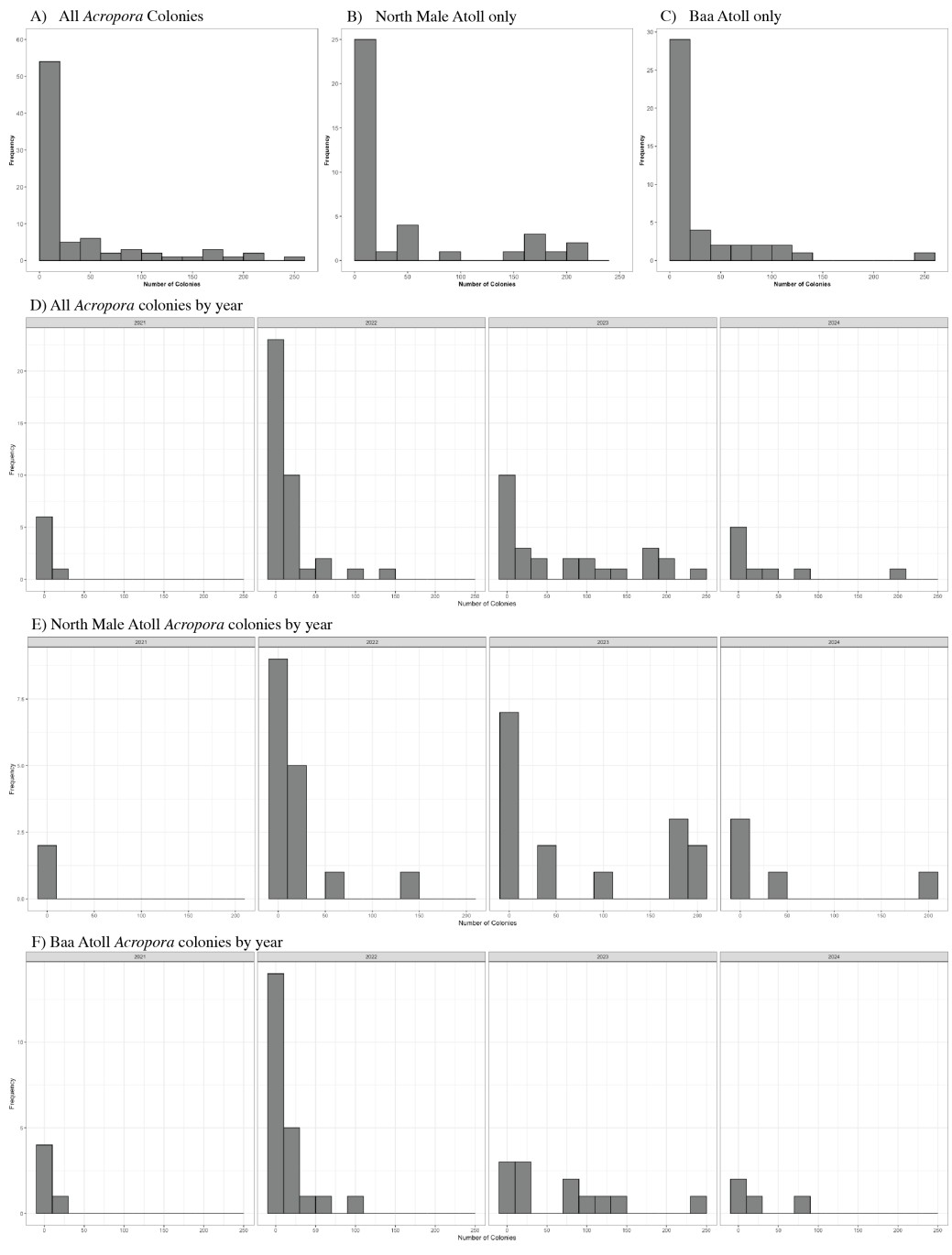

**Figure 4** **Histograms to show the frequency of the number of colonies (count) to spawn per day.** (A) Includes data from all *Acropora* colonies ($N = 3{,}026$). (B) Includes data from North Male Atoll only ($N = 1{,}709$). (C) Includes data from Baa Atoll only ($N = 1{,}317$). (D) Displays data from all *Acropora* colonies by year. Note data collection in 2021 ($N = 40$) and 2024 ($N = 364$) include one spawning season, and data from 2022 ($N = 652$) and 2023 ($N = 1{,}970$) include two spawning seasons. (E) Displays data from North Male Atoll *Acropora* colonies by year. 

**Figure 4 (...continued)**
Note data collection in 2021 ($N = 8$) and 2024 ($N = 250$) include one spawning season, and data from 2022 ($N = 311$) and 2023 ($N = 1,140$) include two spawning seasons. (F) Displays data from Baa Atoll *Acropora* colonies by year. Note data collection in 2021 ($N = 32$) and 2024 ($N = 114$) include one spawning season, and data from 2022 ($N = 341$) and 2023 ($N = 830$) include two spawning seasons.

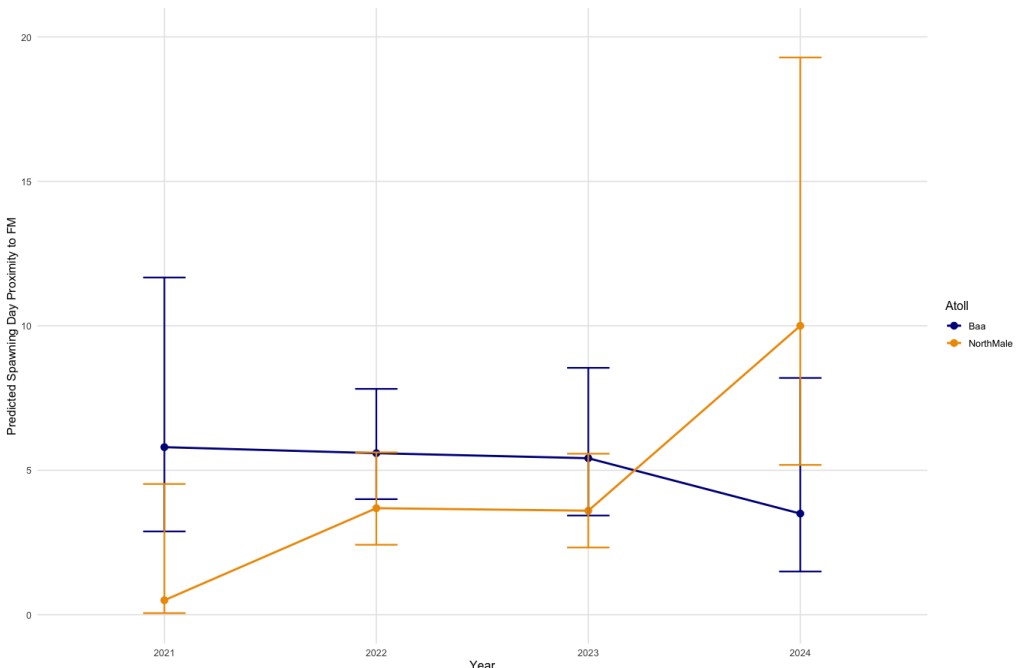

**Figure 5** Effect of year on spawning day deviation from the full moon, for each atoll.

found that a higher proportion of colonies spawn following higher levels of precipitation (Table 1; $p = 0.038$). This relationship was also significant when looking at the proportion of *A. secale* colonies to spawn per day ($p < 0.001$). Additionally, a higher proportion of *Acropora* colonies spawn closer to the full moon ($p = 0.042$). No other environmental conditions were statistically significant predictors of the proportion of colonies to spawn on a given day for *A. secale, A. tenuis, A. humilis* or *Acropora* at the genus level (Table 1).

## Effect of local environmental conditions on spawning day

Spawning events of *Acropora* corals closer to the full moon are significantly correlated with lower tide depths across both atolls (GLM: SE = 1.465, $Z = 2.304$, $p = 0.021$; Table 2). The effect of the average SST 30 days prior to spawning was dependent on atoll: in Baa Atoll it was not a significant effect but in North Male, spawning events closer to the full moon were significantly correlated with lower average SST 30 days prior to spawning (GLM: SE = 0.266, $Z = 2.057$, $p = 0.040$; Table 2). Average daily wind speeds and total daily precipitation were not significant predictors of spawning day relative to the full moon in either atoll.
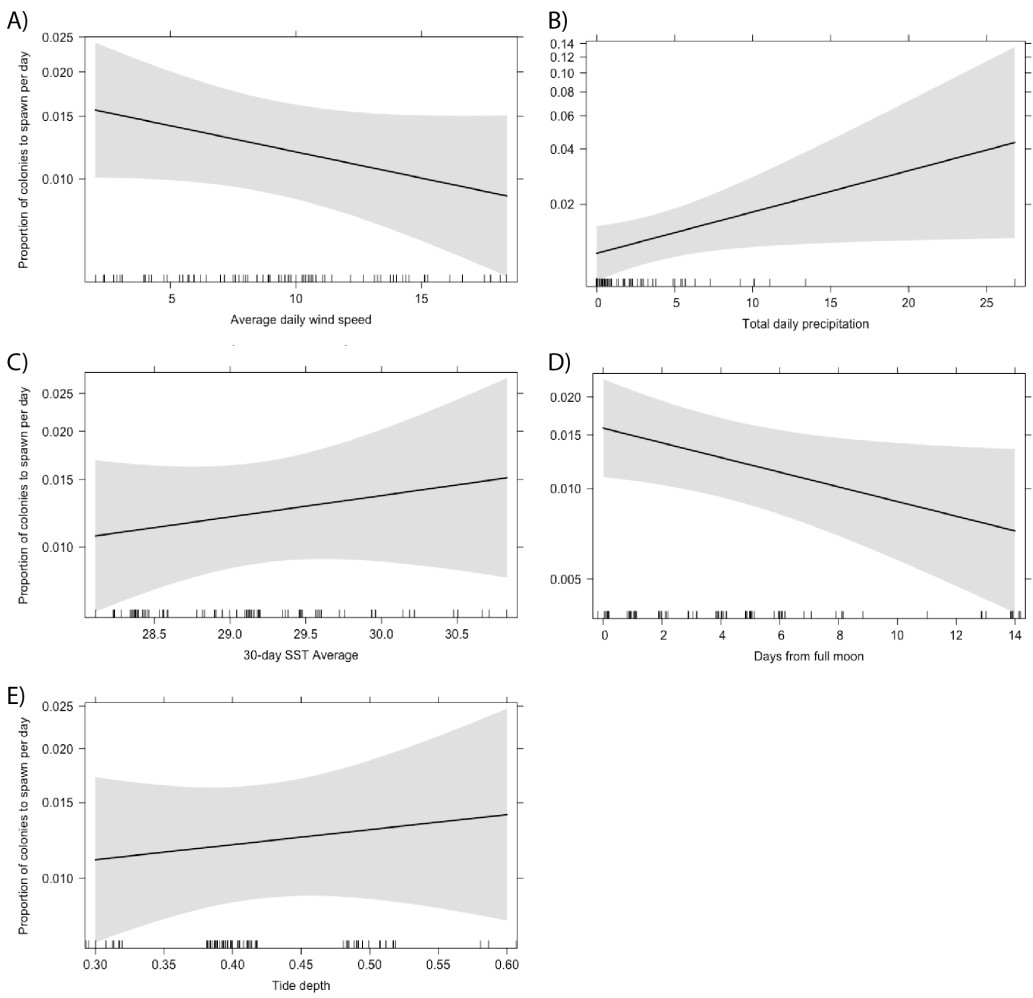

**Figure 6** **Effect of environmental conditions on the proportion of *Acropora* colonies to spawn per day based on the results of a generalised linear model fit with a beta distribution.** The explanatory variables included in the model were average daily wind speed (A), total daily precipitation (B), the average sea surface temperature (SST) for a 30-day period prior to the spawn day (C), the proximity of the spawn day to the nearest full moon (D), and the tide depth at the lowest tide (E).

Similarly, tide depth is a significant predictor of the spawning proximity to the full moon for *A. secale* and *A. humilis*. However, for *A. secale* the relationship was variable depending on Atoll. In Baa Atoll, there was a significant, negative relationship, but a significantly positive relationship in North Male Atoll (Table 2). While both are significant, the effect of the relationship is much stronger in North Male, indicating tide depth has a greater influence there. Spawning events of *A. secale* and *A. humiils* were also significantly correlated with lower levels of daily precipitation (Table 2), indicating some species-specific relationships between precipitation and spawning day. Additionally, for *A. humilis* the effect of average daily wind speeds is different between the two atolls, which in North Male demonstrates a significant, positive relationship (Table 2). No predictors were significant when modelling the effect of environmental conditions on *A. tenuis*.

**Table 1   Results of generalised linear models to explore the relationship between the degree of spawning synchronicity of *Acropora* species and environmental conditions.** Significant results are given in bold. Values less than 0.001 (highly significant) are denoted as < 0.001. Species were chosen for species specific modelling based where $N > 30$ ($N$ = number of days in which spawning was recorded). Proportion of colonies spawning per day (response) was calculated as the number of colonies spawning each day divided by the total number of colonies recorded of each species. All models were fitted with a beta distribution.

| Response | Species | Explanatory: fixed effects | Estimated coefficient | SE | Z-value | p-value |
|---|---|---|---|---|---|---|
| Proportion of colonies spawning per day | *All Acropora* ($N = 81$) | Average daily wind speed | −0.034 | 0.023 | −1.502 | 0.133 |
| | | Total daily precipitation | 0.053 | 0.026 | 2.075 | **0.038** |
| | | Average SST for prior 30 days | 0.130 | 0.159 | 0.813 | 0.416 |
| | | Tide depth at lowest tide | 0.813 | 1.351 | 0.601 | 0.548 |
| | | Proximity to full moon (day) | −0.056 | 0.028 | −2.039 | **0.042** |
| | | Marginal $R^2$ | 0.033 | | | |
| | | Residual Deviance | −560.808 | | | |
| | | Log likelihood | 280.402 | | | |
| | *A secale* ($N = 36$) | Average daily wind speed | −0.030 | 0.043 | −0.697 | 0.486 |
| | | Total daily precipitation | 0.099 | 0.022 | 4.552 | **<0.001** |
| | | Average SST for prior 30 days | 0.257 | 0.434 | 0.593 | 0.553 |
| | | Tide depth at lowest tide | 0.752 | 1.603 | 0.469 | 0.639 |
| | | Proximity to full moon (day) | −0.054 | 0.050 | −1.078 | 0.281 |
| | | Marginal $R^2$ | 0.103 | | | |
| | | Residual Deviance | −203.809 | | | |
| | | Log likelihood | 101.904 | | | |
| | *A tenuis* ($N = 41$) | Average daily wind speed | −0.043 | 0.039 | −1.092 | 0.275 |
| | | Total daily precipitation | −0.009 | 0.043 | −0.213 | 0.832 |
| | | Average SST for prior 30 days | 0.060 | 0.206 | 0.292 | 0.770 |
| | | Tide depth at lowest tide | 0.116 | 1.924 | 0.060 | 0.952 |
| | | Proximity to full moon (day) | −0.052 | 0.032 | −1.641 | 0.101 |
| | | Marginal $R^2$ | 0.030 | | | |
| | | Residual Deviance | −231.204 | | | |
| | | Log likelihood | 115.602 | | | |
| | *A humilis* ($N = 30$) | Average daily wind speed | −0.074 | 0.032 | −2.340 | **0.019** |
| | | Total daily precipitation | 0.019 | 0.057 | 0.329 | 0.742 |
| | | Average SST for prior 30 days | 0.282 | 0.219 | 1.286 | 0.199 |
| | | Tide depth at lowest tide | 0.772 | 1.781 | 0.433 | 0.665 |
| | | Proximity to full moon (day) | 0.117 | 0.091 | 1.282 | 0.200 |
| | | Marginal $R^2$ | 0.081 | | | |
| | | Residual Deviance | −152.971 | | | |
| | | Log likelihood | 76.485 | | | |

# DISCUSSION

The effect of local environmental conditions on the timing and synchronicity of coral spawning in *Acropora* species in the Maldives is highly variable between factors, spatially, and temporally. We show the relationships and potential for environmental conditions to act as triggers for coral spawning patterns in the Maldives are complex and provide the first

**Table 2  Results of generalised linear models to explore the relationship between the spawning day deviation from full moon of *Acropora*** species and environmental conditions, and their interactions with Atoll. Significant results are given in bold. Values less than 0.001 (highly significant) are denoted as <0.001. Species were chosen for species specific modelling based where $N > 30$ ($N$ = number of days in which spawning was recorded).

| Response | Species | Explanatory: fixed effects | Estimated coefficient | SE | Z-value | p-value |
|---|---|---|---|---|---|---|
| Spawning day proximity to the nearest full moon | *All Acropora* ($N = 81$) | Average daily wind speed | 0.006 | 0.039 | 0.154 | 0.878 |
| | | Atoll | −17.096 | 8.236 | −2.076 | **0.038** |
| | | Total daily precipitation | 0.003 | 0.040 | 0.067 | 0.946 |
| | | Average SST for prior 30 days | −0.086 | 0.180 | −0.478 | 0.633 |
| | | Tide depth at lowest tide | 3.375 | 1.465 | 2.304 | **0.021** |
| | | Average daily wind speed*Atoll | 0.032 | 0.048 | 0.657 | 0.512 |
| | | Total daily precipitation*Atoll | 0.031 | 0.044 | 0.713 | 0.476 |
| | | Average SST for prior 30 days*Atoll | 0.547 | 0.266 | 2.057 | **0.040** |
| | | Tide depth at lowest tide*Atoll | 1.727 | 2.866 | 0.603 | 0.547 |
| | | Marginal $R^2$ | 0.323 | | | |
| | | Residual Deviance | 385.2 | | | |
| | *A. secale* ($N = 36$) | Average daily wind speed | 0.024 | 0.036 | 0.681 | 0.496 |
| | | Atoll | −6.540 | 1.316 | −4.970 | **<0.001** |
| | | Total daily precipitation | 0.054 | 0.015 | 3.695 | **<0.001** |
| | | Average SST for prior 30 days | −0.224 | 0.238 | −0.942 | 0.346 |
| | | Tide depth at lowest tide | −2.803 | 0.686 | −4.084 | **<0.001** |
| | | Average daily wind speed*Atoll | −0.090 | 0.047 | −1.915 | 0.056 |
| | | Tide depth at lowest tide*Atoll | 16.522 | 3.493 | 4.730 | **<0.001** |
| | | Marginal $R^2$ | 0.742 | | | |
| | | Residual Deviance | 145.961 | | | |
| | *A. tenuis* ($N = 41$) | Average daily wind speed | 0.034 | 0.045 | 0.758 | 0.448 |
| | | Total daily precipitation | 0.054 | 0.054 | 0.998 | 0.318 |
| | | Average SST for prior 30 days | 0.262 | 0.235 | 1.116 | 0.265 |
| | | Tide depth at lowest tide | −1.979 | 1.852 | −1.069 | 0.285 |
| | | Nagelkerke's $R^2$ | 0.138 | | | |
| | | Residual Deviance | 209.871 | | | |
| | *A. humilis* ($N = 30$) | Average daily wind speed | −0.071 | 0.049 | −1.462 | 0.144 |
| | | Atoll | −1.729 | 0.656 | −2.635 | **0.008** |
| | | Total daily precipitation | 0.101 | 0.041 | 2.456 | **0.014** |
| | | Average SST for prior 30 days | −0.163 | 0.174 | −0.941 | 0.347 |
| | | Tide depth at lowest tide | 5.267 | 2.455 | 2.145 | **0.032** |
| | | Average daily wind speed*Atoll | 0.172 | 0.084 | 2.052 | **0.040** |
| | | Marginal $R^2$ | 0.516 | | | |
| | | Residual Deviance | 100.959 | | | |

analysis from the region of spawning synchronicity. Furthermore, we identified that these relationships are variable between species within the *Acropora* genus. While we provide valuable insights into how environmental factors might influence the observed variability in spawning patterns, we must also conclude there are other unexplored influences or a broad

spectrum of ecologically suitable conditions for spawning. However, our results highlight the critical role of environmental factors in predicting spawning events of *Acropora* corals, given the observed variations in timing and synchronicity both within and across species.

*Acropora* corals in the Maldives spawn during synchronised multispecific spawning events more frequently and in greater numbers than during single species spawning events, challenging the hypothesis of a breakdown in spawning synchrony in equatorial regions (*Oliver et al., 1988*). Other studies have challenged the equatorial asynchronous breeding hypothesis by documenting spawning synchrony, including in Palau (*Penland et al., 2004*; *Gouezo et al., 2020*), the Solomon islands among 28 *Acropora* species (*Baird, Marshall & Wolstenholme, 2002*), and in Singapore where 18 species from ten genera were recorded spawning over three nights in March 2002 (*Guest et al., 2005a*). Recent research based on 90 sites throughout the Indo-Pacific found no correlation between latitude and *Acropora* reproductive synchrony at the lunar month level, but that reproductive synchrony was highly variable temporally and spatially (*Bouwmeester et al., 2021*). However, reports from Kenya show patterns of asynchrony in *Acropora* spawning (*Mangubhai & Harrison, 2008*). The presence of synchronised multispecific spawning on some equatorial reefs, but not others suggest latitude cannot be used to infer spawning synchrony, but that synchronised multispecific spawning is also not a characteristic of equatorial reefs. Overall, our findings support the argument that no marine environment is aseasonal and that reproductive seasonality, synchronised multispecific spawning, and mass spawning events, can be features of equatorial reefs (*Guest et al., 2005a*), as there are clear peak months of spawning for *Acropora* corals in the Maldives.

On days in which multispecific or conspecific spawning of *Acropora* corals occurred, the number of colonies to spawn was highly varied. We sought to examine whether this variability can be explained by variability in environmental conditions between days. Total daily precipitation was a significant predictor, though the relationship was unexpected; we found that increased precipitation was positively correlated with the number of colonies to spawn each night. We had predicted that fewer colonies would spawn on days with higher rainfall as decreased salinity can negatively affect fertilisation (*Scott, Harrison & Brooks, 2013*). However, rainfall occurring several hours prior to spawning events may have minimal impact on fertilisation and so the resolution of our data could be obscuring the true effect, which might be more evident on a finer temporal scale, such as hourly. The presence of species-specific relationships could further support this interpretation given precipitation was also a significant predictor of the proportion of *A. secale* colonies to spawn per day, and the spawning day of *A. secale* and *A. humilis* relative to the full moon, but not a significant predictor of either response variable of *A. tenuis*—a species known to release gametes earlier on the night of spawning than other species in the genus (*Harrison et al., 1984*; *Hayashibara et al., 1993*; *Fukami et al., 2003*; *Monfared et al., 2023*). It is also plausible that increased precipitation acts as a stressor to the corals (*Haapkylä et al., 2011*), causing colonies to release their gametes and thus we observed higher numbers of colonies spawning on nights of increased rainfall. Alternatively, our results may be demonstrative of spawning only occurring during times of low precipitation, since our models only included days upon which spawning was observed and therefore do not account for variation in

environmental conditions during days with no spawning. A range of zero to 26.8 mm of precipitation was recorded during spawning days, but 1,683 colonies (55.6%) spawned on days with less than one mm of rainfall, further supporting this interpretation. Therefore, while a greater proportion of colonies per day spawned on nights with higher rainfall, more colonies in total spawned on nights with lower rainfall.

In general, for *Acropora* at the genus and species level, the local environmental cues included in our models were not strong predictors of the proportion of colonies to spawn on a given day. It has been hypothesised that a narrower variation in environmental parameters near the equator results in more constant conditions, prompting marine organisms to exhibit a more protracted reproductive season or the ability to breed year round (*Orton, 1920*; *Giese & Pearse, 1974*; *Oliver et al., 1988*; *Mangubhai & Harrison, 2008*). This narrow variation makes the ability for any one local cue to act as a primary driver of spawning less probable and thus may explain our results, including the low range of precipitation recorded on spawning days. It is possible the environmental conditions observed on spawning days already fall within the optimal conditions for *Acropora* spawning, supporting the equatorial protracted breeding season hypothesis (*Oliver et al., 1988*).

While synchrony was observed in Maldivian *Acropora* corals, the degree of synchrony between species was more variable. Only ~58% of the recorded spawning days featured more than one species, but over 96% of all recorded colonies spawned during multispecific spawning events. When looking at the species level, some temporal reproductive isolation in relation to the night of spawning is evident. For example, on 29.3% of the days in which *A. tenuis* spawned it was the only species to do so. Similar patterns of temporal reproductive isolation have been observed in *Acropora* species in Kenya, Australia, and the Red Sea (*Shlesinger & Loya, 1985*; *Mangubhai & Harrison, 2008*; *Gilmour et al., 2016*). Temporal reproductive isolation may contribute to higher coral cover and fertilisation success by reducing the likelihood of interbreeding and reducing competition for resources and recruitment spaces (*Gilmour, Speed & Babcock, 2016*; *Shlesinger & Loya, 2019*). However researchers have begun to question whether high coral cover is indicative of healthy and functional communities as a breakdown in synchrony or extended spawning periods could reduce settlement and result in aging coral populations (*Shlesinger & Loya, 2019*). Overall, our study expands our knowledge of *Acropora* spawning synchrony on Maldivian reefs, but the region remains distinctly understudied. The extent to which coral cover and subsequent reef health is regulated by observed synchrony and breeding seasons remains largely unexplored, and the degree of synchrony between *Acropora* corals and other genera, or within other genera, is currently unknown.

When considering a monthly temporal scale, we found that a change in SST does not determine the likelihood of *Acropora* spawning to occur in a given month, but it does significantly predict the number of *Acropora* colonies to spawn per month. Considering our use of binary logistic regressions, this is unsurprising, as protracted spawning seasons in the Maldives with two distinct peak spawning months results in months where few colonies are observed spawning (*e.g.*, January, $N = 1$) are weighted equally to peak months (*e.g.*, November, $N = 1,891$). However, understanding that a rapid rise in SST can predict the
number of colonies to spawn is beneficial in predicting mass spawning events. Rising SSTs have been known to coincide with major spawning events in other tropical and equatorial reefs, which has been hypothesised to be driven by peaks of insolation (*Penland et al., 2004*; *Permata et al., 2012*; *Jamodiong et al., 2018*). When looking at individual species, spawning of *A. humilis* is more likely to occur during months following a rise in SST. However, a change in SST was not a significant predictor of the probability of *A. secale* or *A. tenuis* spawning in a given month, or for *Acropora* corals as a whole. *A. humilis* was recorded spawning in five months, of which four have been identified as the peak spawning months in the Maldives (*Monfared et al., 2023*). Thus, we can infer from these models that a rapid rise in SST can predict peak spawning months, but not necessarily the months where a small number of colonies spawn (*e.g.*, less than 10). Our results support findings from a range of latitudes in the Indo-Pacific that a rise in SST acts as a driver of spawning, to maximise gamete density following the final stages of corals' gametogenic cycle (*Mangubhai & Harrison, 2008*; *Keith et al., 2016*), and contradicts observations on Palauan reefs that found changes in SST were not significant predictors of spawning (*Penland et al., 2004*; *Gouezo et al., 2020*) and the lack of correlation between *Acropora* reproductive synchrony and latitude documented by *Bouwmeester et al. (2021)*. Rising ocean temperatures could be disrupting gametogenic and spawning cycles (*Shlesinger & Loya, 2019*). Thus, understanding the relationship between SSTs and monthly coral spawning patterns in the Maldives will be beneficial for the management of these reef systems due to increased reliance on targeted conservation measures to retain genetic diversity and re-populate degraded reefs, such as *in-situ* larval settlement which relies on the collection of diverse genotypes from a sufficient number of donor colonies. Furthermore, understanding the existing relationship between rising ocean temperatures and marine organisms' reproductive cycles, coupled with the degree of synchrony and seasonality within breeding cycles, will be pivotal to comprehensively assess the impacts of a changing climate on such ecosystems.

On a finer temporal scale, the mean SST in the month prior to spawning has no effect on spawning day relative to the full moon for *A. secale*, *A. tenuis*, or *A. humilis*. Given the effect of SST on monthly spawning times is likely linked to promoting the final stages of gametogenesis (*Nozawa, 2012*; *Keith et al., 2016*), it seems following this development other environmental factors have a greater influence on determining the exact night of spawning. However, in North Male only, for all *Acropora* colonies, spawning closer to the full moon is correlated with lower SSTs in the 30 days prior to spawning. In North Male only, spawning was also documented further from the full moon in 2024 compared to other years. In April 2024, the National Oceanic and Atmospheric Administration (NOAA) and International Coral Reef Initiative (ICRI) confirmed the fourth global bleaching event (*National Oceanic and Atmospheric Administration, 2024*). At the time of writing, the extent of the impact of this bleaching event on Maldivian reefs is unknown, but based on informal documentations online and in the grey literature, it is considered to be widespread with high levels of mortality observed in *Acropora* corals (*Goreau, 2024*; *Shahid, 2024*). *Acropora* corals in Baa Atoll consistently spawned further from the full moon compared to North Male Atoll, and thus in 2024 there was no difference observed. Overall, North Male shows a delay in spawning day with higher mean SSTs 30 days prior to spawning, which is consistent

with a delay in spawning during a global bleaching event. Research on *A. hyacinthus* from the Philippines found lower fecundity in some colonies following exposure to bleaching and other environmental stressors, leading colonies to potentially skip a spawning season (*Jamodiong et al., 2018*). Bleaching can have adverse and long-lasting effects on coral reproduction, including declines in egg count and reproductive polyps in bleached or previously bleached colonies, and a loss of unique genotypes following periods of mass mortality (*Ward, Harrison & Hoegh-Guldberg, 2000*; *Levitan et al., 2014*). A study from Scott Reef, an isolated oceanic reef system in northwestern Australia, found early recovery to be driven by the survival and growth of remnant colonies, and subsequent recovery and juvenile recruitment was reliant on the reproductive output of the local, surviving colonies (*Gilmour et al., 2013*). Although Maldivian atolls form part of an extensive, interconnected reef system, recovery from a global bleaching event which impacted the entire atoll chain will be reliant on surviving local populations retaining genetic diversity and reproductive capacity. A global bleaching event underscores the urgency of understanding reproductive patterns of coral species on temporal and spatial scales to elucidate our knowledge on the mechanisms driving coral reef decline and the potential pathways for recovery. A comprehensive understanding of coral spawning dynamics pre- and post-bleaching events over time enables the development of targeted conservation measures aimed at supporting coral reproduction and enhancing the resilience of coral reef ecosystems in the face of ongoing threats and the triple planetary crisis. In the aftermath of a global bleaching event, where conservation of coral reefs will be reliant on retaining and enhancing genetic diversity, further research on how the bleaching event impacted the observed relationships between environmental conditions and *Acropora* spawning patterns in the Maldives will be vital to accurately predict spawning timings and synchronicity.

Although multiple synchronised multispecific spawning events were observed in the Maldives, spawning was observed over multiple months, lunar phases, and days within each lunar phase. It has been hypothesised that variability in spawning days within lunar cycles may be influenced by cues from lunar rhythms. *Wolstenholme et al. (2018)* proposed the third lunar quarter synchronises with zero declination, the point in which the moon is directly aligned with the Earth's equatorial plane, once or occasionally twice, per year. They suggested that this coincidence of lunar factors influences the timing of annual mass coral spawning and may be associated with low atmospheric pressure. Our findings do somewhat support this interpretation, as two thirds of observed colonies did spawn during the third quarter (*i.e.*, zero-to-six days after the full moon). Furthermore, studies of *A. millepora* on the Great Barrier Reef found moonlight to be an important external stimulus for mass spawning synchrony in relation to the time of night of gamete release (*Kaniewska et al., 2015*). However, we observed spawning over several lunar phases, which is consistent with observations of *Acropora corals* in other reefs around the world (*e.g.*, *Hayashibara et al., 1993*; *Nozawa, 2012*; *Lin & Nozawa, 2017*; *Komoto et al., 2023*). Further evidence that the lunar phase does not predict spawning time comes from Caribbean reefs, where spawning of *Acropora* corals typically occurs two-to-six days after the full moon, but have been recorded closer to the full moon and around the new moon (*Fogarty, Vollmer & Levitan, 2012*). Additionally, in Kenya, *Acropora* corals have been observed spawning across

multiple lunar phases and during both spring and neap tides, with the majority of colonies spawning after the full moon and new moon (*Mangubhai & Harrison, 2008*). Similarly in the Maldives spawning was recorded during spring and neap tides, but 676 colonies spawned between one and two days before the full moon, compared to only two in Kenya during this lunar period (*Mangubhai & Harrison, 2008*). Overall, while *Acropora* corals do spawn over multiple lunar phases in the Maldives, spawning is concentrated around the full moon, with 96.3% of colonies spawning between two days before and seven days after the full moon.

We observed differences in spawning day between atolls for *Acropora* at the genus level, and in *A. secale* and *A. humilis*. This difference underscores the need for further research from additional locations to fully understand the extent of spawning patterns spatially and temporally in the Maldives. The proportion of colonies to spawn per day is consistent across atolls, which is to be expected assuming similar reef compositions between atolls in the Maldives. We identified that spawning day closer to the full moon is significantly correlated with lower tide depths, which is unsurprising given the association between full moons and spring tides (*Reis-Filho et al., 2011*), and therefore is unlikely to explain inter-atoll differences. When considering the *Acropora* genus, the insignificant effect of wind speed and daily precipitation coupled with an $R^2$ of 0.323, along with the limited effect of environmental conditions on the proportion of colonies to spawn per day, suggests there are additional factors influencing spawning patterns and perhaps explaining inter-atoll differences. Despite documenting spawning across multiple lunar phases, moonlight is known to regulate spawning timings on a hourly temporal scale and corals rely on natural light cycles to regulate numerous physiological, biological and behavioural processes (*Kaniewska et al., 2015*; *Ayalon et al., 2021*). An analysis of a global dataset on coral spawning found corals exposed to light pollution spawned between one and three days closer to the full moon than those on unlit reefs (*Davies et al., 2023*), which could explain the inter-atoll differences in spawning day in the Maldives (*Monfared et al., 2023*). Studies have demonstrated a loss of synchrony occurring in high- and low-light environments (*Shlesinger & Loya, 2019*), and that light pollution delayed gametogenesis and unsynchronised gamete release in two *Acropora* species in the Indo-Pacific (*Ayalon et al., 2021*). Experiments on colonies of *A. millepora* exposed to ambient, artificial, and no light conditions found only corals exposed to ambient light spawned at the expected time (of night), with artificial light disrupting spawning and no spawning occurring in the colonies in the dark (*Kaniewska et al., 2015*). Overall, these studies provide evidence that light pollution has the potential to disrupt reproductive timing, and the need to protect reefs from artificial light. Coupled with rising ocean temperatures, there is potential for significant impacts from anthropogenic activity on corals' reproductive patterns, underscoring the importance of documenting baseline information of the phenology of reef systems. Given our results indicate additional factors may be influencing spawning patterns on Maldivian reefs, the effect of light conditions and light pollution should be explored.

This study presents the first species-specific analysis from the Maldives of *Acropora* spawning patterns. At the individual species level, the environmental variables examined

were generally weak predictors of synchronicity and varied in their effect on spawning day. Different species also exhibit distinct spawning patterns between atolls and throughout the year, in the months in which they spawn and their peak spawning months. Variability in spawning timing between species, in relation to spawning month, day, and the precise timing of gamete release at night, has been widely documented in the *Acropora* genus (*Harrison et al., 1984*; *Van Oppen et al., 2002*; *Fukami et al., 2003*; *Van Woesik, 2010*; *Levitan et al., 2014*; *Monfared et al., 2023*). In this study, most spawning by *A. secale* was recorded in North Male Atoll and only occurred between October and December. While similar numbers of *A. humilis* colonies spawned in each atoll, those in North Male Atoll predominately spawned earlier in the year whereas the majority in Baa Atoll spawned later. In contrast, *A. tenuis* spawned only during the second spawning season in Baa Atoll, but over five months in North Male Atoll. These patterns suggest that species-specific responses to environmental cues may differ given the natural variations in environmental conditions throughout the year and between atolls, even in equatorial regions. Spatial and temporal separation likely results in exposure to varying environmental conditions, which may influence reproductive timing and synchronicity. Protracted breeding seasons may also promote genetic divergence, as documented in western Australia where colonies of *A. tenuis* that spawned during different seasons separated into two genetically distinct groups (*Gilmour, Speed & Babcock, 2016*). Although all three species examined in our study spawned over five months, only *A. tenuis* did so within a single atoll. Its' broad spawning duration, couple with potential exposure to a higher variability in environmental conditions, may reflect underlying genetic divergence or adaptive plasticity. Further research of the genetic structure of *Acropora* populations in the Maldives is needed to test these hypotheses. Finally, variations in species' cue hierarchies may lead to differing responses to environmental factors.

Following a mass bleaching event, coral conservation in the Maldives is likely to focus on targeted *in-situ* larval settlement in order to re-populate reefs and enhance genetic diversity. We found a significant difference between days in the number of colonies to spawn, with a range of one colony to over 200 colonies, but a strong preference of *Acropora* colonies to spawn during multispecific spawning events. Given this preference, the inclusion of the genus level models also holds value in understanding the predictors of spawning events and levels of synchronicity. Furthermore, dependence on *in-situ* recruitment following mass mortality across Maldivians reefs underscores the importance of obtaining exact spawning day observations within each lunar phase, as well as understanding spawning patterns at the species level. However, whether environmental conditions can cause or prevent spawning to occur on a given day remains unexplored. By using a binary logistic regression to account for days without spawning, future research could also investigate the potential for changes in weather to drive spawning events. It is also evident the explanatory variables considered in this study as predictors for spawning day do not account for the full pattern of the data, and thus exploring additional cues of spawning in the Maldives would be beneficial. Finally, ground truthing the satellite derived measurements used in this study, using *in-situ* measurements of environmental conditions, would also be beneficial to further explore the

potential for such conditions to act as triggers for spawning timing and synchronicity of Maldivian corals, and due to the spatial variability of spawning occurrence in the Maldives.

## CONCLUSION

Our data demonstrates that local environmental conditions are influencing the timing of *Acropora* corals spawning in the Maldives, across varying temporal scales, expanding our understanding of reef ecology in a historically understudied region. The relationship between environmental conditions and degree of synchronicity within *Acropora* corals is variable between species, but whether environmental conditions can prevent, or cause spawning remains unexplored. The data presented in this study also provides a 3-year baseline into *Acropora* spawning patterns and their relationship with local environmental conditions prior to a global bleaching event, providing crucial insights beneficial in aiding the recovery of Maldivian reefs. Understanding these baseline patterns of reproductive phenology of Maldivian reefs will be vital to understand the extent to which anthropogenic pressures and rising ocean temperatures alter such patterns, and how to best address these problems. It is evident there is a need for further research into intra- and inter-atoll coral reproductive phenology in the Maldives, as well as explorations of additional genera spawning patterns, to fully understand seasonal variations in coral spawning patterns, and how these could be impacted by a changing climate.

## ACKNOWLEDGEMENTS

This research took place at Four Seasons Landaa Giraavaru, Baa Atoll and The Sheraton Full Moon Resort and Spa, Furana Fushi, North Male Atoll. We would like to thank both properties for their continued and unwavering support throughout this study. We would also like to thank the wider Reefscapers team who assisted during data collection throughout the study period, with particular thanks to Katelyn Hegarty-Kelly, Laura Alonso Diaz, Matthew Gledhill, and Akbar Ahmed.

### Funding
The authors received no funding for this work.

### Competing Interests
Kate Sheridan, Margaux Monfared, Simon Dixon and Amelia Errington were employed by Reefscapers at various stages of this research. Thomas Le Berre is managing director of Reefscapers. Margaux A.A. Monfared is employed by Blue Pangolin Consulting Ltd and Simon P. Dixon is employed by Coral Vita.

### Author Contributions
- Kate Sheridan conceived and designed the experiments, performed the experiments, analyzed the data, prepared figures and/or tables, authored or reviewed drafts of the article, and approved the final draft.

- Margaux A.A. Monfared conceived and designed the experiments, performed the experiments, authored or reviewed drafts of the article, and approved the final draft.
- Simon P. Dixon conceived and designed the experiments, performed the experiments, authored or reviewed drafts of the article, and approved the final draft.
- Amelia J.F. Errington conceived and designed the experiments, performed the experiments, authored or reviewed drafts of the article, and approved the final draft.
- Thomas Le Berre conceived and designed the experiments, authored or reviewed drafts of the article, and approved the final draft.

## Data Availability

The raw dataset and the code used to conduct analysis are available in the Supplementary Files.

## Supplemental Information

Supplemental information for this article can be found online at http://dx.doi.org/10.7717/peerj.19447#supplemental-information.

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
