# Peer review of "Synchrony on the reef: how environmental factors shape coral spawning patterns in Acropora corals in the Maldives"

_PeerJ, doi:10.7717/peerj.19447_

## Round 0.1 · original submission · Major Revisions

Two expert reviewers have evaluated your manuscript and have positive comments aboujt the work that you have done but also have provided excellent comments and suggestions to improve the manuscript.

The manuscript provides very useful data on coral spawning in an area that has not been extensively studied. Please ensure that you pay attention to the reviewers comments in particular with respect to clarifying the metholodolgy that was used. Also the introduction and discussion can be enhanced by including more literature to improve the context and scope fo the manuscript.

Please ensure that you address all of the issues raised by both reviewers in a detailed rebuttal letter, where you clearly indicate what changes have been made and where so that the revision process runs smoothly.

Reviewer 1 ·

Basic reporting

This manuscript investigates Acropora spawning timing in the Maldives and explores how environmental factors influence spawning patterns, including timing and synchronization. Overall, the manuscript is well-written. However, it shows significant overlap in both data and research direction with Monfared et al. (2023), as both studies aim to examine the relationship between environmental factors and coral spawning timing. While this study includes additional data and analysis, I strongly recommend revising the manuscript to eliminate overlapping content and focus on the novel aspects of this research. Specifically, highlighting how environmental factors influence coral spawning synchronization would enhance the originality and contribution of this study to the field.

Experimental design

After reviewing the raw data, I noticed that there are several spawning days where only a single colony spawned. From an ecological perspective, spawning by a single colony is unlikely to contribute significantly to the coral population's offspring. Therefore, I recommend excluding these data points and focusing only on days when more than two colonies spawned. After this adjustment, you may observe clearer peaks in spawning days per month at each site. Using these more ecologically meaningful "mature" data for analysis could potentially yield more impactful and interpretable results.

Validity of the findings

It is recommended to include the estimated coefficients for each fixed effect variable along with their corresponding 95% confidence intervals. This will help readers better understand the direction and magnitude of the variables' effects. Also, adding the Residual Deviance in the results to provide more information about the model's goodness of fit.

Presenting the main results in graphical form could greatly enhance readers' understanding of the findings. Referring to the methods used in Keith et al. (2016) and Sakai et al. (2020), visual representation can more intuitively illustrate the influence of environmental factors on coral spawning.

Additional comments

Introduction:
* Line 75: It appears that the research by “Shlesinger & Loya” was not conducted in equatorial regions. Please verify and revise this statement as necessary.
* Lines 116–121: The citations “van Woesik, 2010; Keith et al. 2016” in this paragraph appear to discuss the correlation between spawning months and wind speed (strong and weak correlations, respectively), not spawning days. Additionally, Sakai et al., 2020 suggests that wind speed 1–2 months before spawning correlates with deviations in spawning days. These points should not be conflated; I recommend clarifying them for accuracy.
Figures:
* Fig. 1a: Simplify the latitude and longitude representation; such a high level of precision is unnecessary.
* Fig. 4: It is unclear why the y-axis is unevenly scaled. Including a label for 0 (FM) might improve readability. Additionally, the two site-specific graphs could be combined into a single graph, using different colors to distinguish between sites. This would streamline the presentation.

Reviewer 2 ·

Basic reporting

General comments:
While the manuscript is well-written, several sentences and paragraphs could be improved and streamlined. Most references are current, but there are notable gaps in citing foundational literature, which would enrich the manuscript and may provide a more critical interpretation and discussion of the data. Additionally, the broader ecological and conservation implications of your findings could be emphasized more prominently, particularly in the context of increasing marine heatwaves.
Consider rephrasing some sentences and paragraphs in the introduction and discussion to avoid redundancy, particularly regarding the importance of spawning synchrony and environmental factors. Instead, I would recommend providing more comparisons with studies from other regions and species.
Consider standardizing and explaining terminology better when discussing synchronicity and spawning timing. I also think that the discussion of climate-related stressors, such as marine heatwaves, on coral reproduction could be expanded.

Specific comments:
Line 25 – Although mentioned later in the text, it is worthwhile clarifying here why these three species were chosen.
Line 28 – I would recommend replacing “test” with “estimate”.
Line 57 – There is no doubt that Babcock et al. (1986) is a key paper that should be discussed, however, it is not the first to document broadcast-spawning as stated. On a broad statement like that, I would recommend including references to several of the pioneering works in the field such as Szmant-Froelich et al. 1980 Biology Bulletin, Kojis and Quinn 1981 Bulletin of Marine Science, Kojis and Quinn 1981 Marine Ecology Progress Series, Fadlallah and Pearse 1982 Marine Biology, Harriott 1983 Coral reefs, Krupp 1983 Coral Reefs, Babcock 1984 Coral Reefs, Harrison et al. 1984 Science, Shlesinger and Loya 1985 Science, Wallace 1985 Marine biology, Szmant 1986 Coral Reefs. Particularly relevant to your manuscript are those papers that studied multiple species or that focused on Acropora like Wallace 1985.
Line 61 – Consider adding and discussing Gilmour et al. (2013) Recovery of an isolated coral reef system following severe disturbance. Science.
Line 65 – I recommend changing the ambiguous term ‘health’ to ‘state’.
Line 74 – Reframe the equatorial topic as recent studies challenge earlier assumptions and also some of the references provided did not take place in equatorial reefs. While I think it is ok to mention this idea in the following sentences, I suggest that lines 73-75 simply focus on asynchronous spawning and extended reproductive periods. You can include additional works like Shlesinger et al. 1998 Marine Biology, Penland et al. 2004 Coral Reefs, Gilmour et al. 2016 Plos one, Jamodiong et al. 2018 Invertebrate reproduction & development; Zoological studies.
Lines 81-91 – Regarding bi-annual cycles, I suggest having a look at some of Rosser and Gilmour papers from Western Australia. As for differences in month of spawning and environmental factors I suggest also referring to van Woesik et al. 2006 Ecology letters, van Woesik 2010 Proceedings B, Howells et al. 2014 Scientific reports, Keith et al. 2016 Proceedings B.

Experimental design

General comments:
The methods section requires more precise details about sampling and nightly surveys (e.g., specific dates, number of samples, lunar phases, and survey frequency).
Adding supplementary materials, such as raw data visualizations, could enhance clarity regarding both methodological aspects and results.
I would also recommend providing more detail on the statistical diagnostics performed to validate the GLMs.
Moreover, although the authors acknowledge nicely in several places some possible limitations of the data, I think that there is room to expand on potential limitations associated, for example, with using satellite-derived environmental data, especially for precipitation and SST, and how these might influence the findings. Accordingly, the interpretation of some findings warrants further consideration.
The lack of significant predictors for some species (e.g., A. tenuis) could be addressed in more depth, considering potential methodological or ecological explanations.

Specific comments:
Line 200 – “focal general”? Did the authors mean “focal genera”?
Lines 211-212 – Provide exact details of nightly spawning surveys (e.g., total nights per month/season/year, Lunar phases, etc.) rather than simply stating September 2023 – April 2024. Can we be certain that the choice of surveys’ timing does not bias the conclusions about spawning near full moons (e.g., were there in situ surveys during other lunar phases?). Is it possible that there might be spawning events in other lunar phases, but no surveys were performed to detect them?
Lines 248-251 Please clarify better how spawning proportions were measured or calculated. Provide specifics on colony count and their temporal resolution.
Line 253 – Seems like “Atoll” should be treated as a fixed factor in the analysis.
Lines 276-277 – Perhaps finer temporal scales for temperature effects should be considered. Although focusing on octocorals, see for example Liberman et al. 2022 Ecology.
Lines 278-279 – For some more comparisons of lunar days and spawning, I suggest having a look at Wolstenholme et al. 2018 Invertebrate reproduction & development (among others), while also noticing that many spawning events do not occur on full moons but also on other lunar phases.

Validity of the findings

Line 315 – Please clarify in the main text – did the authors perform in situ spawning surveys over eight months or many of these reports are based on the classifications of egg maturity from sampled colonies? Are the surveys performed on specific lunar phases or during the entire month? Please also provide detailed information regarding the exact dates and numbers of sampled corals.
Lines 393 – 407 – Another plausible hypothesis could be that these Acropora corals had mature gametes that were ready to be released within the right period (of days to weeks) and increased precipitation during this period acted as a stressor causing corals to release their gametes.
Lines 408 – 428 – But see contrasting conclusions by Bouwmeester et al. 2021 Coral reefs. I think it should be discussed.
Lines 489 – 525 – The authors have here three summarizing paragraphs, all starting with “This study”. I suggest rewriting these and making them more concise and less repetitive. Instead, I would recommend adding more depth to the discussion by adding and discussing more comparisons with studies on extended coral spawning periods and/or asynchronous spawning patterns. Also, since the authors emphasize the valid point of the possible deleterious effects of climate change, global warming, and coral bleaching on coral reproduction and therefore why it is important to study it, I suggest adding discussion on some of the many papers describing detailed studies of coral reproduction and spawning before and after bleaching.

Comments regarding the figures:
Figure 1 – Align panel letter “A)” to the left like in all other figure panels.
Figure 2 – Beautiful figure! For accuracy, the gametes shown in panel 2a are not ‘bundled’ yet (it only occurs while they start being pushed out during spawning).
Figure 3 – Since most observations include 0-50 colonies spawning per day perhaps it is best to adjust the histogram bin sizes for more nuanced patterns. Also, it might be worth adding a supplementary set of histograms by year.
Figure 4 – Improve clarity and labeling. For example, the y-axis values are not spread uniformly and missing labels across the axis, so it is hard to understand what the values and scales are below 5. Also, consider raw data histograms or barplots for spawning day frequencies.

Additional comments

Dear Authors,
Your manuscript presents an interesting study on the timing, synchronicity, and environmental drivers of Acropora spawning events. I commend your extensive monitoring efforts and recognize the additional year of observations as a valuable contribution, alongside the new analyses provided. These additions differentiate this manuscript from your earlier work and enhance its potential for publication. However, as specified above, I believe significant revisions are necessary before the manuscript can be published.
In summary, I appreciate the effort invested in this study and its potential to advance our knowledge of coral reproduction. I believe that addressing the points above will substantially strengthen the manuscript and its relevance, and I hope my comments will aid in doing so.

---

## Round 0.2 · Major Revisions

I have received the evaluation of your resubmission from one reviewer and agree with their comments and suggestions. Please ensure that in your resubmission that your rebuttal letter and your manuscript are congruent and that you attend to all of the reviewers´ comments from this and from the previous evaluations. Clearly note these in your rebuttal letter.

Reviewer 2 ·

Basic reporting

The authors have addressed many of my previous comments and revised the manuscript accordingly. While these revisions have led to some improvements, it appears that the manuscript has undergone only a minor revision rather than the major revision that was requested. Several key concerns remain insufficiently addressed. For example, my previous review recommended providing more detailed methodological information, but this has not been fully incorporated in the current revision. More concerningly, there are inconsistencies between the response letter and the actual manuscript. Some revisions mentioned in the response letter do not appear in the text, while others differ significantly from what was described. Two specific examples highlight this issue:
In response to a comment on line 25 of the abstract, the authors stated that they had added a sentence, but the sentence they reference does not match what was actually included.
In the revised manuscript, the caption of Figure 2 does not reflect the changes the authors claim to have made.
There are likely additional discrepancies, and I strongly encourage the authors to carefully cross-check their responses with the revised text to ensure that all revisions have been accurately implemented. Changes described in the response letter should be clearly reflected in the manuscript in the same exact language, with precise indications of where they appear.
Furthermore, I recommend incorporating additional relevant literature on Acropora corals’ extended spawning periods across different lunar phases, which is more appropriate than many of the currently cited papers. These papers include tedious work and compilations by different groups that should be acknowledged for their effort and contributions. Specifically, I suggest discussing more in depth some of the findings in: Mangubhai and Harrison (2008, MEPS), Fogarty et al. (2012, PLOS ONE), Jamodiong et al. (2018, Zoological Studies and Marine Biodiversity), Shlesinger and Loya (2019, Science), among others.
Additionally, the potential role of light pollution in shaping differences in spawning timing and synchrony, particularly in atolls, warrants discussion. The following studies provide valuable insights on this topic: Kaniewska et al. (2015, eLife), Ayalon et al. (2021, Current Biology)
Finally, the references require thorough corrections. Inconsistent formatting remains an issue, with capitalization varying between titles, genera and species names not italicized, and some references containing missing information or typographical errors.
Given these concerns, I believe the manuscript still requires a major revision before it can be considered for publication. I encourage the authors to carefully revise their work, ensuring that all changes are accurately reflected and that attention is given to both substantive content and finer details.

Experimental design

no comment

Validity of the findings

no comment

---

## Round 0.3 · accepted · Accept

An expert reviewer who read your resubmission and I are satisfied with the changes that you have made to the manuscript. I am recommending that the manuscript be accepted for publication in PeerJ. Congratulations.

Reviewer 2 ·

Basic reporting

No further comments

Experimental design

No further comments

Validity of the findings

No further comments